# Colour Preference of the Deer Ked *Lipoptena Fortisetosa* (Diptera: Hippoboscidae)

**DOI:** 10.3390/insects12090845

**Published:** 2021-09-19

**Authors:** Annalisa Andreani, Marzia Cristiana Rosi, Roberto Guidi, David Jafrancesco, Alessandro Farini, Antonio Belcari, Patrizia Sacchetti

**Affiliations:** 1Department of Agriculture, Food, Environment and Forestry (DAGRI), University of Florence, Piazzale delle Cascine 18, 50144 Firenze, Italy; annalisa.andreani@unifi.it (A.A.); marziacristiana.rosi@unifi.it (M.C.R.); roberto.guidi@unifi.it (R.G.); antonio.belcari@unifi.it (A.B.); 2National Institute of Optics (Istituto Nazionale di Ottica, CNR-INO), Largo Fermi 6, 50125 Firenze, Italy; david.jafrancesco@ino.cnr.it (D.J.); alessandro.farini@ino.cnr.it (A.F.)

**Keywords:** chromotropic traps, visual stimuli, colour attraction, hippoboscid flies, capture pattern

## Abstract

**Simple Summary:**

Insects use visual stimuli to find habitats, food, or a mate while moving around. This trait might be exploited to intercept flying insects to monitor their populations and reduce their presence. Among the various visual stimuli, colours are commonly used to attract insects. *Lipoptena fortisetosa* is a hematophagous deer ectoparasite native to Japan that has spread to several central European countries and was recently recorded in Italy. Measures to monitor and control *L. fortisetosa* would be helpful given its potential threat as a pathogen vector for animals and humans. The objective of this research was to assess the potential use of colour to attract and trap *L. fortisetosa*. The response of the winged adults was evaluated through an experimental trial carried out in a wooded area of Tuscany using differently coloured sticky panels as traps. Blue panels attracted the highest number while yellow panels showed the lowest performance. This preference for blue could be useful in the design of traps to reduce the population of this parasitic fly which, at certain times, can reach a very high density, causing annoyance to wildlife and humans visiting natural areas.

**Abstract:**

*Lipoptena fortisetosa*, a deer ked native to Japan, has established itself in several European countries and was recently recorded in Italy. This hippoboscid ectoparasite can develop high density populations, causing annoyance to animals and concern regarding the potential risk of transmitting pathogens to humans. No monitoring or control methods for *L. fortisetosa* have been applied or tested up to now. This research evaluated the possible response of *L. fortisetosa* winged adults to different colours as the basis for a monitoring and control strategy. In the summer of 2020, a series of six differently coloured sticky panels were randomly set as traps in a wooded area used by deer for resting. The results indicated a clear preference of the deer ked for the blue panels that caught the highest number of flies during the experimental period. Lower numbers of flies were trapped on the red, green, black, and white panels, with the yellow panels recording the fewest captures. The response clearly demonstrates that this species displays a colour preference, and that coloured traps might be useful for monitoring and limiting this biting ectoparasite in natural areas harbouring wildlife and visited by people.

## 1. Introduction

*Lipoptena fortisetosa* is a small blood-sucking ectoparasite whose primary hosts are various species of ruminant artiodactyl mammals, especially cervids and bovids [1], although it is also known to bite humans. Phylogenetically, it belongs to the subfamily Lipopteninae, but unlike other hippoboscids, it is not able to frequently switch host and attaches itself to a single animal for life. Newly emerged *Lipoptena* flies are fully winged adults that immediately search for a suitable host. Once found, the fly settles down to live in the host’s fur and gradually loses its wings, which separate at a predetermined break line on the proximal part of the wing [2,3,4].

*Lipoptena fortisetosa* is native to Japan but has spread to many European countries [5], including Italy, where it was recorded for the first time in 2019 in wooded areas in Tuscany [4]. *L. fortisetosa* is an obligate, permanent ectoparasite which, apart from its original host, sika deer, thrives on a limited range of closely related mammalian species, especially cervids [6,7,8,9]. Before settling on the so called “definitive host”, this fly may occasionally bite other species for food, including humans [10,11].

In general, parasites are either dependent on a single species or have adapted to a wider range of hosts. The level of host selectivity varies greatly among parasites and affects the degree of specificity of the host choice process, that in some cases needs to be highly precise since any host may not provide all the requirements necessary for a parasite’s survival. The closer the parasite’s association with a few species, the deeper its level of adaptation; consequently, exploiting other species becomes more difficult [12]. The host location is determined by many factors that may act in concert, such as insect morphology, physiology, behaviour, ecology, genetics, and circadian cycle. These are distinct, evolved responses that represent adaptations to specific biotic and abiotic constraints [12,13]. This process is affected by many issues, including habitat, movement, odour, and moisture, and is crucial for ensuring the parasite’s survival [14]. For blood-sucking insects, the search for a host can be divided into three phases that are not strictly consecutive: a) appetitive searching; b) activation and orientation; and c) attraction [12,14]. In general, the location of the host involves a variety of chemical, physical, and visual signals, such as specific odours emitted by animals, carbon dioxide, movements, and the shape and colour of the host [13]. Usually, visual and olfactory stimuli act over a long distance, while humidity and heat are more significant at closer range [12].

In the superfamily Hippoboscoidea, some species belonging to Nycteriibidae and Streblidae were studied in terms of host location [15,16], but a substantial amount of information is available especially on other Hippoboscoidea, such as tsetse flies (Glossinidae) due to their great economic, medical, and veterinary importance [17,18]. Host finding by Glossinidae was found to entail two kinds of behavioural responses: long-range olfactory responses and short-range responses, determined by olfactory and visual factors [13]. Visual stimuli are of primary importance at short distances: past experiments on tsetse flies demonstrated that, in a series of tested colours, phthalogen blue traps had a significantly higher capture rate than those of other colours, such as yellow [19].

Host location behaviour has been poorly investigated in members of the Hippoboscidae family; in fact, except for earlier observations reported by Bequaert [20], the only experimental research has been carried out in Finland, where the preferences of *L. cervi* for host body parts, colour, and temperature were investigated using people as dummies. Among other findings, the winged adults displayed a clear attraction toward people wearing dark and red clothing [21].

*Lipoptena fortisetosa* has never been studied for its host location process although it is currently receiving renewed attention, especially given its medical and veterinary importance. In fact, as ascertained in other hippoboscid species, *L. fortisetosa* may be a potential vector of pathogens that are harmful for animals and humans [22,23,24,25,26]. Since this parasite lives on just a few host species, completes its life cycle while dwelling permanently on a single subject, and is not able to frequently switch victims, we believe it has developed efficient mechanisms to locate a host after emergence.

Investigations into two hippoboscid flies, *Hippobosca equina* and *L. cervi*, demonstrated that visual signals are involved in host location and that mainly colour stimuli are used [21,27]. Thus, evaluating the colour preferences of *L. fortisetosa* might be useful for disclosing behavioural traits of this allochthonous ectoparasite which is spreading through Europe, causing concern for its medical and veterinary importance. Moreover, a possible response to colours could be exploited to define monitoring and control strategies. In fact, coloured traps coated with odourless glue are frequently used to sample different blood-feeding insects since they are inexpensive and easy to assemble [28].

The objective of this paper is to provide a fresh account on the response of *L. fortisetosa* to visual stimuli through an experiment carried out in a wooded area of Tuscany using differently coloured sticky panels as traps.

## 2. Materials and Methods

The field trial was conducted in a wooded area in Schignano (Prato, Tuscany, central Italy) at about 550 m a.s.l. (43.967432 N; 11.101761 E), where many warnings about the abundance of ked flies have been reported by people visiting this area. The study area consisted of a sloped clearing enclosed on three sides by a forest of mainly oak, holm oak, and chestnut that is frequently used by deer as a passageway or rest area.

### 2.1. Experimental Design

To evaluate the possible response of ked flies to colour, three series of differently coloured sticky traps were arranged in three different locations within the experimental area (Figure 1). The first series bordered the forest to the southwest of the clearing and had an east–west orientation (trap sides with exposure north–south); the second was placed inside the woods and was oriented north–south (trap sides with east–west exposure); and the third series was placed to the north of the glade with the same orientation and trap exposure as the second series.

Each series consisted of three repetitions of six solid colours (black, transparent, blue, green, yellow, and red) in a randomized sequence for a total of 54 traps (18 traps per series) (Figure 2). The colours were chosen on the basis of studies conducted on tsetse flies and other hematophagous insects [18,21,29,30,31]. The transparent colour was used as a control. Coloured traps consisted of plastic alveolar polypropylene “plastonda” panels while transparent traps were made of Poliver (artificial glass polystyrene); all of them measured 20 cm × 30 cm × 2.5 mm (width, height, and thickness, respectively) and were purchased at the home improvement retailer OBI Italia. The spectral reflectance of these panels in the visible and UV regions, between 250 and 800 nm, was measured using a PerkinElmer Lambda 1050 spectrophotometer coupled with a specific accessory for reflectance measurements (150 mm InGaAs Integrating Sphere) (Perkin Elmer Inc., Waltham, MA, USA). The black and transparent traps showed nearly constant reflectance over the wavelength range from 250 to 800 nm. The black colour exhibited a mean reflectance of about 6%, while for the transparent sample it was 17%. The blue trap displayed a maximum reflectance of 420–470 nm (∼60% of reflectance) while the green displayed 500–550 nm (∼20% of reflectance). The yellow panel showed a maximum reflectance of 520–550 nm (∼75% of reflectance), and the red, 620–650 nm (∼55% of reflectance) (Figure 3).

Both sides of the panels were coated with a glue applied by brush (Planatol VP 1854 PSA, Ivog biotechnical systems GmbH, Neüsaß, Germany). In each series, traps were arranged 1.5 m above the ground and hung from two cords in a stationary position; the 18 panels within each series were placed close to each other. To avoid any interference from the juxtaposition of colours, the order was changed every two weeks according to a randomized experimental design.

The traps were set on 6 July 2020 and checked weekly: flies caught on each side were counted separately for every panel and then removed to estimate their response to colours. The traps remained continuously exposed until 29 October 2020, and this time period was chosen based on *L. fortisetosa* adults in Europe being reported as present from June to October [5].

On the same day as the trap control, sweeping paths were performed in different environments near the experimental area (woodland, forest edge, open field, track).

### 2.2. Statistical Analyses

Analyses were carried out considering the mean number of flies caught by each trap per day as a dependent variable and the following as independent variables: the position of the series (position 1, position 2, and position 3), trap colour, sampling date, and trap activity (dichotomised as traps that captured *L. fortisetosa* or did not). From 1 October onwards, most of the traps were inactive and an extremely low number of flies (9) was caught. For this reason, only data from 15 July to 1 October were analysed.

To highlight potential differences in trap attractiveness, the data structure was checked through factor analysis of mixed data (FAMD), and then inferential statistics were applied.

The goal of the FAMD analysis was to explore the association between all the variables and highlight which factors determined the variability of the average number of flies caught by the traps. FAMD was performed with the open-source software RStudio (RStudio Version 1.3.1093, 2009–2020, PBC, Boston, MA; http://www.rstudio.com/) using the packages FactoMineR and FactoExtra. All variables were considered active, and missing data were managed using the package missMDA [32,33].

After the FAMD observation, inferential analysis was carried out to highlight differences among active traps compared to inactive traps as well as the average number of caught flies.

The proportion of active with respect to inactive traps was analysed using a Chi-square test (6 × 3 contingency tables) for the null hypothesis that all proportions were equal. When necessary, multiple pairwise comparisons were applied to the number of active traps according to the variable series position (H_0_: p_1_ = p_2_; H_0_: p_1_ = p_3_; and H_0_: p_2_ = p_3_) and to the variable trap colour (H_0_: p_1_ = p_2_ = … = p_6_). The Type I error rate was adjusted, thereby reducing the maximum error rate of 0.05 by the total number of comparisons [34]. Chi-square tests were carried out using Excel software (Microsoft 365, 2016, Microsoft Italia, Milano, Italy).

Differences in the average number of *L. fortisetosa* captured on traps of different colours were analysed using univariate analysis of variance with the sampling dates from 15 July to 1 October and panel colours as factors, while the dependent variable was the average number of *L. fortisetosa* captured per day. Data were log(x + 1) transformed [34], and a pairwise comparison was performed using the Bonferroni test if a main effect was highlighted. All statistical analyses were performed using IBM SPSS Statistics for Windows, Version 25.0 (IBM Corp, Armonk, NY, USA) software.

## 3. Results

### 3.1. Overall Evaluation of the Experiment Set on Lipoptena fortisetosa Captures

The eigenvalue to explain the total variance of data was selected according to the criteria defined by Karlis et al., [35]. Dimension 1 (Dim. 1) satisfied the criterion (eigenvalue >1.31) and summarized most information; however, to interpret the FAMD results, the second dimension (Dim. 2) was also considered. Both dimensions explained 17.04% of the overall variability. Although it was less than 50%, the representation was valid for describing the variance distribution within the dataset. The results of the FAMD analysis are reported in Table 1.

Figure 4a shows that the two variables of number of captures and trap activity were linked to the first dimension. Total variability was mainly explained by the number of captured *L. fortisetosa* (Contr. = 33.93%), and these observations were positively correlated with the first dimension (Corr. = 85.70%). The variable trap activity also contributed to the construction of Dim. 1 (Contr. = 27.91%) and was positive correlated to active traps (Corr. = 1.39) as opposed to the inactive traps (Corr. = –1.05) (Figure 4b). Observations that positively correlated to the Dim. 1 may be characterized by a higher average number of captures per trap.

In addition, the variable position of a series was related to Dim. 1 (Figure 4a), with position 1 contributing to the first component structure (Contr. = 16.07%, Corr. = 1.52). Position 1 was opposed to position 3 as evidenced by its high contribution and significant positive ratio compared to Dim. 2 (Contr. = 15.65, Corr. = 0.73) (Figure 4b and Table 1).

The variable sampling date was less correlated with Dim. 1 than with Dim. 2 (Figure 4a); only three dates correlated with Dim. 1, with 12 and 20 August being positive (Corr. = 0.96 and 1.45, respectively), and 1 October being negatively correlated (Corr. = –0.96).

Regarding the variable colour, Dim. 1 and Dim. 2 contrasted blue traps with yellow ones (Figure 4b). The blue traps were correlated with both dimensions (Dim. 1: Corr. = 0.65, Dim. 2: Corr. = 0.59) while the yellow ones were negatively correlated (Dim. 1: Corr. = –0.48, Dim. 2: Corr. = –0.38).

The FAMD results emphasized that position 1 was more appropriate for insect monitoring since it explained most of the variability and was linked to active traps. Moreover, 12 and 20 August seemed to be the most favourable days for *L. fortisetosa* capture. The outcomes suggest that there were optimal capturing periods that should be taken into account. The traps with the most divergent results were the blue and the yellow ones, with the blue traps being the most active.

### 3.2. Trap Activity

A significantly different number of active traps was highlighted between the three positions of the series (χ^2^ = 19.93, df = 10, *p* = 0.03). Pairwise comparisons showed a significantly higher number of active traps in position 1 (71.48%) compared to position 2 (37.50%) (χ^2^ = 20.88, df=5, *p* < 0.001) and position 3 (20.37%) (χ^2^ = 15.85, df = 5, *p* = 0.007). The number of active traps in position 3, where the lowest values were recorded, was comparable to the number of active traps in position 2 (χ^2^ = 9.74, df = 5, *p* = 0.08). Considering the variable colour, there was a similar proportion of active compared to inactive traps (χ^2^ = 11.23, df = 5, *p* = 0.34) (Figure 5).

### 3.3. Colour Preference of Lipoptena fortisetosa Winged Adults

The FAMD and trap activity results show low effectiveness for series positions 2 and 3; thus, only *L. fortisetosa* captured by traps in position 1 were analysed. The log-transformed data of the captured *L. fortisetosa* highlighted significant differences among trap colours (main effect colours F = 15.82, *p* < 0.001) and sampling dates (main effect of the sampling dates F = 43.54, *p* < 0.001). Differences among the average number of flies caught daily by the panels were consistent across all sampling periods (interaction effect F = 1.42, *p* = 0.06). The blue panels recorded the highest number of individuals with an average daily catch of 1.28 ± 1.08 (mean ± SE). On the other hand, the yellow traps showed the lowest average values (0.33 ± 0.29 mean ± SE). The blue and yellow colours were significantly different from all the other panel colours (black, green, red, and transparent), which showed a similar average number of insects (respectively: 0.63 ± 0.58, 0.77 ± 0.72, 0.80 ± 0.88, and 0.64 ± 0.10 mean ± SE) (Figure 6).

Regarding the variable sampling date, the most significant numbers of *L. fortisetosa* were recorded in August (mean ± SE of 1.09 ± 0.16, 1.49 ± 0.21, 1.90 ± 0.21, and 0.82 ± 0.14 respectively for 6, 12, 20, and 26 August). The averages recorded on these days were significantly higher than the values obtained on earlier dates (15–27 July: avg ± SE 0.19 ± 0.03, 0.33 ± 0.06, and 0.34 ± 0.06) and after 26 August. On other days, the averages were statistically similar; however, 9 September (0.68 ± 0.12) was an exception, with a slight increase in the daily catch. This value was similar to those recorded on 6 and 26 August (Figure 7).

In Figure 8, which shows the trend of flies trapped by the blue panels over the whole period, the average number of *L. fortisetosa* was considerably relevant, reaching the highest value of 22 specimens per trap. Moreover, after the August peak, the blue traps also caught flies effectively in September although that peak was less than half (8 insects/trap).

## 4. Discussion

### 4.1. Colour Preference

Colours tested in this study can be qualified or distinguished based on their spectral characteristics: yellow and red fit into the so-called “cut-off” colours, with a steeply sloped spectrum; blue and green relate to the “band reflecting” colours with discrete reflectance peaks; transparent and black, having constant reflectance at any wavelength, can be considered neutral [19,36].

As far as we know, most dipterans possess five types of photoreceptors, which can be sensitive in a wide band of wavelengths from UV up to the green wavelength range [37,38], so we speculated that *L. fortisetosa* followed this general model, although the spectral sensitivity of the deer keds has never been measured.

The capture of *L. fortisetosa* in Schignano (Prato, Italy) using differently coloured traps clearly demonstrates that this parasitic species displays a colour preference, since the winged adults selected colours according to a preference scale. Blue traps caught a significantly higher number of adults, compared to black, red, green, and transparent traps, while yellow traps seemed to be the least attractive. This colour ranking was consistent throughout the experimental period. Colour attractiveness has, to date, been poorly investigated in hippoboscid flies, with the only reported observations on *L. cervi* in Finland [21] where black-coloured clothing worn by moving people was compared to white, blue, and red clothing by evaluating the number of flies that landed on them or were caught by transparent flypaper. In the field trials, deer keds preferred black to white clothing, but showed no significant preferences for blue or red.

Our field trial results on *L. fortisetosa* differ from those of Kortet et al. [21] in that a blue colour preference was determined, though there are differences in the experimental design (coloured plastic panels in a stationary position instead of coloured clothing worn by moving people) and *Lipoptena* species. In the experiment carried out in Finland, other factors influencing host location may have affected deer ked choice, such as movement, chemical cues, and host size. Usually, moving objects are detected by an achromatic photoreceptor channel [39]; however, many insects are attracted by moving objects with particular colours showing the so-called “wavelength-specific behaviour” [40]. Different long-distance stimuli could also have influenced these flies, such as carbon dioxide, or odour cues emitted by the host skin as was demonstrated for different groups of hematophagous dipterans [13]. However, it is difficult to separate the effect of different stimuli and to ascribe, with certainty, the location of a host to movement per se, since a moving subject may also release more carbon dioxide or may be warmer than one that is stationary [41].

In other hematophagous Diptera, such as Glossinidae (tsetse flies), blue proved to be the most attractive colour over a short distance [36]. Tsetse and ked flies show quite similar mouthparts and feeding activity [4,42], but once a suitable host is found they display different parasitic behaviour. Unlike deer keds, *Glossina* species frequently change host.

Additionally, the morphological affinity between these families could also pertain to eye structure and colour preference, which have been thoroughly investigated in tsetse flies [43,44]. In a field trial in Zimbabwe, *Glossina morsitans morsitans* and *G. pallidipes* showed a marked preference for traps covered by royal blue cotton, strongly reflecting blue–green wavelength bands, but poorly reflecting ultraviolet or green–yellow–orange bands [19]. A similar behaviour was described for another important tsetse fly, *Glossina fuscipes fuscipes,* which was particularly attracted by blue cloth panels, especially phthalogen blue panels, while yellow was the least attractive [36]. However, olfactory stimuli can affect the attraction of tsetse flies to different colours [17].

Other flies of economic importance, such as *Stomoxys calcitrans*, were investigated to highlight possible colour preferences that could be used to improve monitoring and control interventions. Experiments with blue and black targets with different patterns demonstrated the importance of blue in attracting flies of this species [45]. In addition, the attraction of *S. calcitrans* to polyethylene blue screens was not affected when different hues of blue were used [31]. Phthalogen blue with spectral sensitivity at 350, 450, and 625 nm markedly influenced *Stomoxys* spp. capture [31,46].

Other blood-sucking dipterans have shown a response to colours partly comparable to that exhibited by *L. fortisetosa*. For example, in Canada, Tabanidae and Simuliidae were tested with differently shaped and coloured traps and generally responded more to colours than to shapes [29]. Remarkably, all the investigated species belonging to both these families were consistently not attracted by yellow silhouettes. Moreover, when subjected to solid blue and solid yellow traps, tabanids were more attracted by the blue ones, and when subjected to blue/yellow striped traps, they always chose the blue part. Similar outcomes were highlighted in other studies [30] all of these tabanid data were quite consistent with our observations on *L. fortisetosa*, especially for the poor attractiveness of yellow. Nevertheless, for some herbivorous dipterans, such as Tephritidae and especially the Mexican fruit fly, *Anastrepha ludens*, laboratory trials showed a particular preference for yellow together with green, over black, red, blue, and white [47]. Similarly, yellow traps are commonly used in monitoring and control strategies in different olive growing areas for the olive fruit fly, *Bactrocera oleae* [48,49]. The recognition of green fruit by fruit flies, as well as the detection of green leaves by other herbivorous insects, assumes an interaction between two receptor types: green-sensitive, which contributes positively, and blue-sensitive, which contributes negatively. The preference displayed by many phytophagous insects for yellow over green stimuli can be explained by this model [50].

A similar mechanism based on receptor opposition was recently proposed for tsetse flies. In these hematophagous insects, the attraction to blue visual bait might be due to an interaction with a positive contribution by a blue-sensitive photoreceptor against photoreceptors sensitive to green–yellow–UV, which contribute negatively [18,36]. Moreover, based on the four sensitivity peaks of the tsetse fly, the number of specimens attracted by different colour panels correlated positively with the blue colour band and with reflectance at 460 nm (blue wavelengths), whereas the correlation was negative in the green colour range and for reflectance at 520 nm (green wavelengths) [36]. Glossinidae and Hippoboscidae share phylogenetic, morphological, and some behavioural features [51,52]; thus, as already supposed, deer keds are thought to display the same visual ecology as seen in the *Glossina* species. As a consequence, catches of *L. fortisetosa* by different coloured panels might have been determined by different reflectance in relation to the sensitivity of fly photoreceptors. The blue panel had a distinct reflectance peak at about 450 nm, which could be in the proximity of the sensitivity peak of *L. fortisetosa*. The green traps showed a reflectance maximum at about 500 and 550 nm, while the yellow ones displayed a “cut off” spectrum between 450 and 520 nm. The lower attractiveness of the yellow panels might be explained by an opposition mechanism of the photoreceptor sensitive to the yellow–green band as proposed for tsetse flies.

In a natural environment, green leaves have a green reflectance peak at 555 nm due to chlorophyll. They contrast with groups of objects that do not reflect the same wavelength (for instance, fruit and flowers). Moreover, grey–red surfaces (such as bark, soil, and animal) are also present, and these have a reflectance that increases gradually as a function of wavelength [53]. For glossinids, it has been suggested that the contrast of blue against the green–yellow reflectance of vegetation could represent a “non-vegetation” stimulus that induces flies to move towards a more feasible stimulus likely coming from a host. Moreover, the stronger attraction for blue could be explained by hypothesizing an important role of the shadow in creating contrasts: the shaded areas where flies rest, or darker areas on the bodies of potential hosts can appear as bluish patches that contrast with the background [18,36].

In different species of Glossinidae, a negative contribution of the green–red and ultraviolet wavelength in attracting tsetse flies was highlighted [18]. In *L. fortisetosa*, the number of individuals caught by green and red traps was higher than for yellow traps but was lower than for blue traps. In the red and green colour panels, however, the number of flies was similar to that found in black and transparent traps. Spectrophotometric analysis of the green panels showed a nearly constant reflectance at different spectrum bands with a maximum 20% at about 500 and 550 nm. On the other hand, the red panels showed a cut-off transition that varied from a minimum value of 535 to a maximum of 635 nm; the lower wavelength chromatic bands were therefore excluded (blue–green region, 410–520 nm).

Because deer have crepuscular and nocturnal activity, they move and find refuge inside the woodland during the day but feed and rest in open areas in the evening and during the night [54,55]. As proposed for the genus *Glossina*, we hypothesized that *L. fortisetosa* flies also show a colour response depending on the perception of the host against the background of green foliage or other background material. Alternatively, winged adults might prefer blue targets because they resemble the shaded areas where hosts rest or the shaded zones of the host body where insects can more easily carry out their parasitic action.

### 4.2. Trap Activity at Different Positions

As reported in Figure 6, the blue traps caught a higher percentage of insects than those of other colours for all positions. Regarding the three tested series positions, we found that hippoboscids were not caught in the same numbers in the different locations. In fact, in position 1, located at the border between the forest and the open area, captures were significantly higher than those in the other two, both of which were positioned among the trees. The more abundant captures of *L. fortisetosa* at the boundary between the woods and clearing were also supported by observations conducted in Japan [56]. These outcomes showed that the environment seemed to affect parasitic attraction, which was higher in position 1 probably due to the ecology of both fly and host. Since *L. fortisetosa* adults spend their entire lives in the fur of their host, reproducing and laying larvae, the pupae are more likely to fall from the host body to the ground in areas where the deer stay and repose. Because newly emerged winged adults are unlikely to be able to fly long distances, as reported for *L. cervi* [2], they are probably more abundant at emerging sites. Moreover, as already shown, the series were differently orientated with the first east–west oriented (trap sides with north–south exposure) and the second and the third north–south oriented (trap sides with east–west exposure). In Glossinidae, as already mentioned, ultraviolet radiation was negatively associated with fly catches [19,36]. Surface UV reflection varies with solar zenith angles and surface type and orientation [57,58]. The different orientation and exposure of the rows in the three positions may have determined a different reflectance that probably induced a lower presence of *L. fortisetosa* in areas with higher ultraviolet reflectivity. This marked difference in captures in the three positions with a total number of 1013 flies in the first compared with the 162 and 77 insects in positions 2 and 3, respectively, may have also been influenced by wind direction and speed. In the end, the flies showed a similar preference to colours in all three series, with blue being the most attractive and yellow the least.

### 4.3. Pattern of Lipoptena fortisetosa Captures in the Sampling Period

Concerning the number of *L. fortisetosa* adults caught during the sampling experiment (from 15 July to 1 October), it is important to highlight the remarkable peak in August, which might be due to adult emergence from pupae laid during the previous year. The slight increase in the number of flies caught in September might be attributed to a second generation from a small number of specimens originating from earlier emerged adults that quickly found a suitable host. Larvae that pupated in early summer could have completed their development and given rise to another emergence of adults favoured by the high temperature. In the Schignano area, the average temperature from mid-July to mid-September was 23.7 °C [59], while it was 14.1 °C from mid-September to the end of October. Lower temperatures of this latter period probably induced pupae to enter diapause until the following year [60]. Our experiment represents the first field-monitoring survey of *L. fortisetosa* and allows us to conclude that it seems to be multivoltine, as has been reported for other European countries [5,61] and for the area of origin [62].

## 5. Conclusions

Our research provides insight into the life cycle and basic visual ecology of *L. fortisetosa*, which may be exploited in the development of strategies for its monitoring and control and opening a new path to acquire fundamental knowledge on these ectoparasites.

Our experiment showed a peak presence of *L. fortisetosa* winged adults in mid-August, suggesting the possibility that, in Italy, this species could produce more than one generation per year.

More interestingly, we provide evidence that *L. fortisetosa* exhibits a preference scale for colours, with blue being the most attractive and yellow the least attractive. The colour ranking displayed by *L. fortisetosa* suggests that this species is able to discriminate colours and uses visual stimuli over short distance. These results could help in the design of traps to monitor and reduce the populations of this parasitic, hematophagous fly which, in some periods, can reach a very high density, causing annoyance by biting humans in natural areas. The way these flies locate a host remains a topic of active investigation, and further observations are needed to better define the complex stimuli that govern this behaviour. In particular, the role played by odours at medium and long distance should be clarified. For instance, we can state that *L. fortisetosa*, contrary to what is reported for *L. cervi*, does not passively search for a host by waiting for an animal to pass by; rather, it actively flies in search of one. This is supported by our capture of winged adults by a sweeping net only when they were flying, and not when they were resting on vegetation.

The noticeable consistency in some aspects of host attraction among different groups of hematophagous insects is surprisingly high and may suggest a general similar set of needs and tendencies in blood-feeding ectoparasites. Particularly, the convergence in visual stimuli and colour preference ranking is evident: dark colours, especially blue, red, and black, are often attractive for parasites. The similar colour preference for different groups of ectoparasites made us wonder why some hematophagous dipterans are particularly attracted by blue and seem to be relatively unattracted by yellow. Future observations on different blue wavelengths and reflectance are needed to set appropriate and effective traps for this parasite.

## Figures and Tables

**Figure 1 insects-12-00845-f001:**
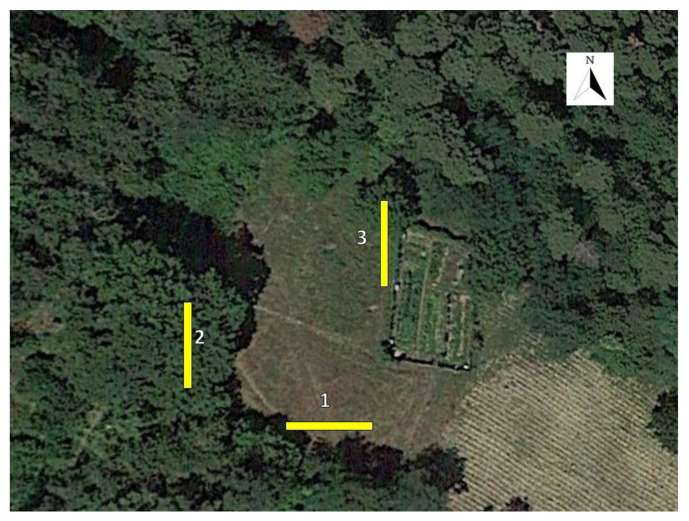
Schignano (43.967432 N; 11.101761 E) (Prato, Italy), 2020. Experimental area with the three series of chromotropic sticky traps arranged for the field trial.

**Figure 2 insects-12-00845-f002:**
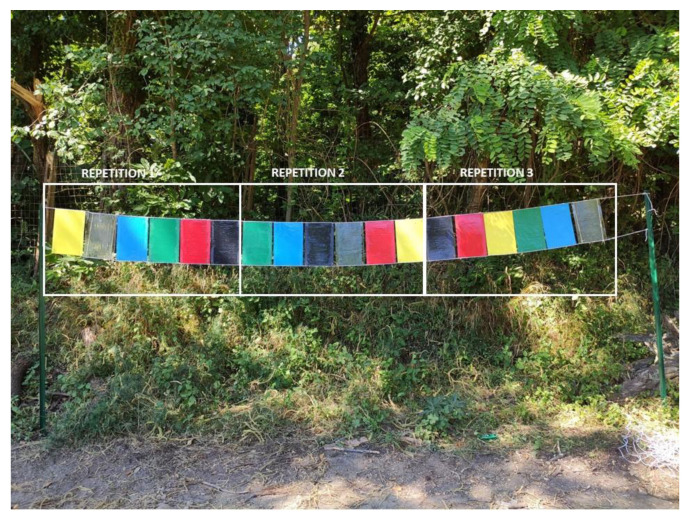
Schignano (Prato, Italy), 2020. Sticky traps of Series 1, formed by three repetitions of six differently coloured plastic panels in a randomized sequence.

**Figure 3 insects-12-00845-f003:**
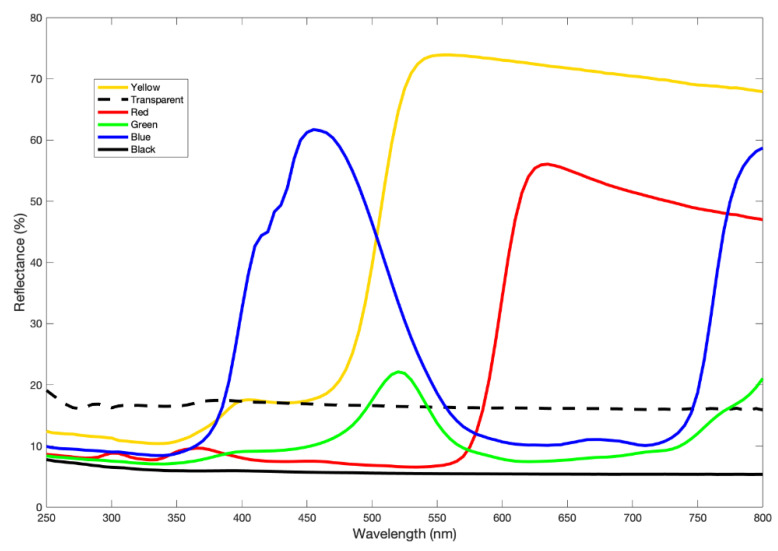
Reflectance spectra of the black (black line), transparent (dotted line), blue (blue line), green (green line), yellow (yellow line), and red (red line) sticky traps exposed to *Lipoptena fortisetosa* winged adults in Schignano (Prato, Italy) between July and October 2020.

**Figure 4 insects-12-00845-f004:**
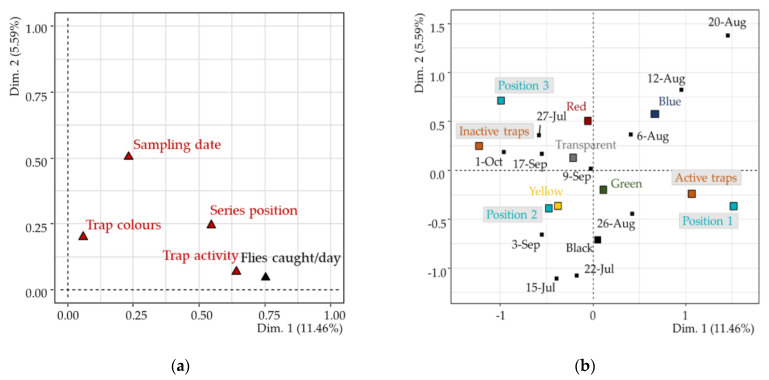
FAMD results: (**a**) Graph of the variables showing the correlation between both quantitative (black triangle) and qualitative variables (red triangles) to Dim. 1 and Dim. 2; (**b**) Graph of the categories: trap activity (active traps and inactive traps), series position (position 1, position 2, and position 3), sampling date (data from 15 July to 1 October), and trap colours (black, transparent, blue, green, yellow, and red). The point for each of the categories indicates the barycentre of the observations.

**Figure 5 insects-12-00845-f005:**
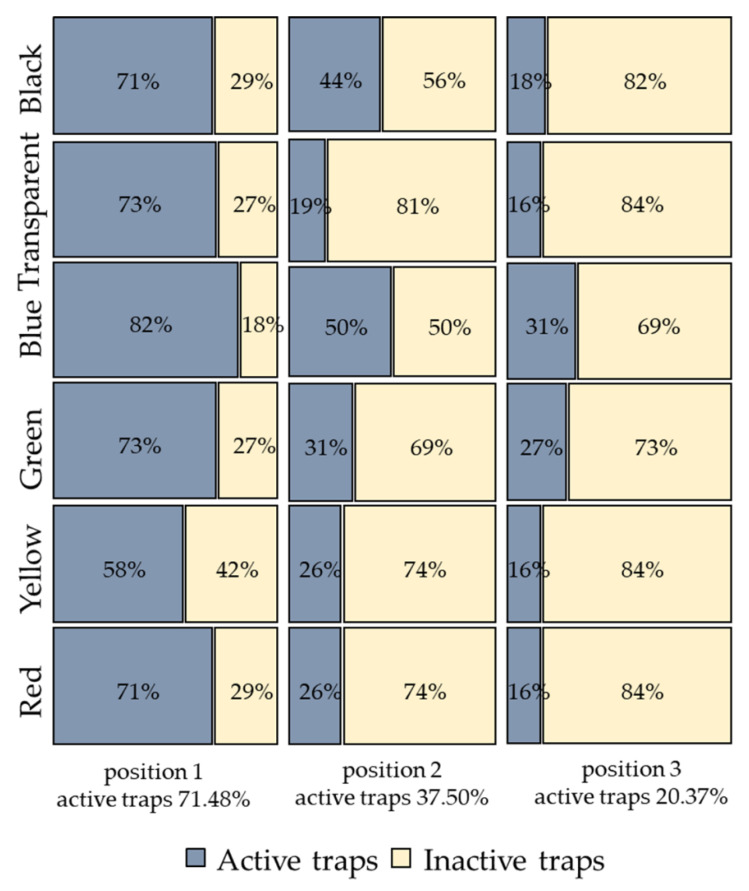
Mosaic plot showing the percentage of active and inactive traps observed in the three series positions. Percentages in the boxes refer to the total number for the colours, while the percentages reported outside the boxes refer to the total number of active traps for each series position.

**Figure 6 insects-12-00845-f006:**
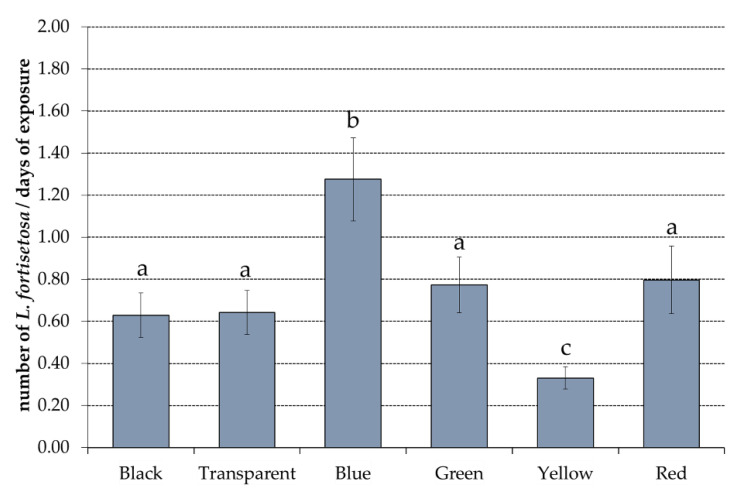
*Lipoptena fortisetosa* captured by panels of six different colours (average calculated on the number of insects of each panel per day of exposure). Different letters above the columns indicate significant differences among colours (main effect colours F = 15.10, *p* < 0.001, followed by multiple comparisons: Bonferroni test, *p* < 0.05).

**Figure 7 insects-12-00845-f007:**
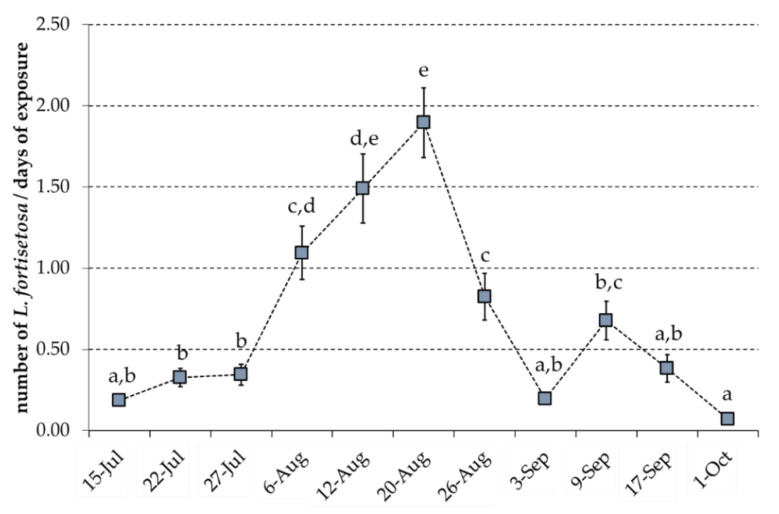
Average number of *Lipoptena fortisetosa* captured by panels on different sampling dates (average calculated from the number of insects caught by each panel per day) in the series position 1. Different letters above the columns indicate significant differences (main effect sampling dates; F = 37.21, *p* < 0.001, followed by multiple comparisons: Bonferroni test, *p* < 0.05).

**Figure 8 insects-12-00845-f008:**
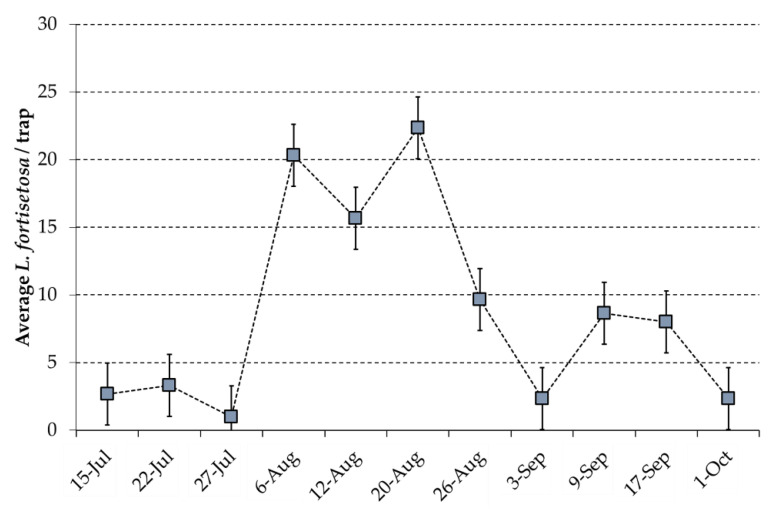
Average number of *Lipoptena fortisetosa* per trap as caught by blue panels from mid-July to 1 October.

**Table 1 insects-12-00845-t001:** Contribution and correlation of the active variables and factors of the categorical variables for the first two principal dimensions of the FAMD.

Variables and Factors	Dim. 1		Dim. 2	
	Contribution	Correlation *	Contribution	Correlation *
**Flies caught/day** ^a^	**33.93**	**0.86**	**4.02**	**0.19**
**Trap activity** ^b^	**27.91**	**0.65**	**5.32**	**0.51**
Active traps	12.75	1.39	2.43	0.24
Inactive traps	1.16	−1.05	2.89	−1.02
**Series position** ^b^	**25.07**	**0.54**	**23.48**	**0.25**
Position 1	16.07	1.51	3.68	−0.35
Position 2	1.72	−0.49	4.15	−0.37
Position 3	7.28	−1.01	15.65	0.73
**Sampling date** ^b^	**10.53**	**0.23**	**47.88**	**0.19**
15 Jul	0.30	−0.39	9.89	−1.10
22 Jul	0.06		9.42	−1.08
27 Jul	0.66	−0.58	1.03	0.36
6 Aug	0.66	0.41	1.09	0.37
12 Aug	0.33	0.96	5.46	0.82
20 Aug	4.09	1.45	15.33	1.37
26 Aug	0.34	0.42	1.61	−0.45
3 Sept	0.59	−0.55	3.55	−0.66
9 Sept	0.00		0.00	
17 Sept	0.60	−0.56	0.23	
1 Oct	1.79	−0.96	0.28	
**Trap colours** ^b^	**2.56**	**0.06**	**19.30**	**0.20**
Black			7.17	−0.69
Transparent	0.19		0.31	
Blue	1.50	0.65	5.17	0.59
Green	0.03		0.50	
Yellow	0.81	−0.48	2.12	−0.38
Red	0.02		4.03	0.52

* Correlation for the variables. Flies caught/day refers to the correlation coefficient, while for the other variables, it refers to the square of the correlation ratio. Correlation was reported when the value was significantly different from 0 (*p* = 0.05). ^a^ continuous variable; ^b^ categorical variable and factors.

## Data Availability

No data were deposited in an official repository.

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
