# Peer review of "Colour Preference of the Deer Ked Lipoptena fortisetosa (Diptera: Hippoboscidae)"

_insects, 2021, doi:10.3390/insects12090845_

Round 1

Reviewer 1 Report

1. In my opinion, the introduction is too extensive and it should be shortened.

2. The material and methods are made acceptable

3. The Results section is acceptable. Dear authors, please specify how far can flies fly? This is important for understanding the distribution of individuals in biotopes. If the traps on the edge and in the clearing are located close, then the flies can simply fly over this distance.

4. In the Conclusion section, you must remove references. It is also necessary to shorten the Conclusion section.

Author Response

Dear Reviewer,

thanks for your comments which allowed us to revise the manuscript.

Please, see our answers in the attached file,

bet regards,

Patrizia Sacchetti

Reviewer 2 Report

The objective of this study is to assess the potential use to color to attract and trap L. fortisetosa, by using different color panels with glued surfaces located in a wooded area. The results indicated that the deer ked preferred the bule panels than the red, green, black and white panels. The experimental designs are proper and the analysis of results clearly shows the color preference and the dynamics of trapped flies during the experimental duration. The results have some scientific merits and the writing is thoroughly clear, therefore I recommend the study to be published.

However, before the study to be published, there are several points should be clarified or improved:

  1. The authors may miss-used the term of ‘color vision’ in this manuscript. For example: lines 34, 310. This study doesn’t do anything to test the deer ked’s color vision but color preference. For testing color vision, by definition of color vision, the authors should do some more tests to demonstrate that the deer ked can choose a particular color regardless the intensity, which has been shown on hone bees by Karl von Frisch in 1914. All the results shown in this study displayed the color preference only.
  2. Materials and Methods 2.1: any measurement of illuminance intensity of the three different locations within the experimental area? The authors should be able to present the differences of the light conditions of the three locations and discuss the effects caused from the differences.
  3. Figure 1: What is the square at the top-right corner for? What are the red circles?
  4. Line 154-156: why two weeks? Any particular reason?
  5. Line 162-163: can’t really understand the description of this sentence for.
  6. Results 3.3: Did authors sort out the trapped L. fortisetosa by sex? How many males or females trapped by the blue panels during the experiments? It would be much better to show whether the blue color traps is sexually or non-sexually attractive to the deer ked.
  7. Line 313-319: the authors compared the color preference with black-colored clothing worn by moving people. It has been know that the visual system detects moving objects with particular channel which is color insensitive. However, some moving object with particular colors may be attraction to some insects, which is so-called wavelength-specific behavior. I suggest the authors to clarify the difference of the color preference and motion detection in this discussion.
  8. The discussion in 4.1 is too lengthy and not focusing. Some of previous studies are not relevant to the study. The authors should be able to make the discussion more concisely.

Author Response

Dear Reviewer 2,

thanks for your stimulating comments which they gave us the opportunity to revise the manuscript.

Please, see the answers to your requests in the attached file,

best regards,

Patrizia Sacchetti
